# Killer-Cell Immunoglobulin-like Receptor Diversity in an Admixed South American Population

**DOI:** 10.3390/cells11182776

**Published:** 2022-09-06

**Authors:** Marlon Castrillon, Nancy D. Marin, Amado J. Karduss-Urueta, Sonia Y. Velasquez, Cristiam M. Alvarez

**Affiliations:** 1Grupo de Inmunología Celular e Inmunogenética (GICIG), Universidad de Antioquia (U-de-A), Medellín 050010, Colombia; 2Division of Oncology, Department of Medicine, Washington University School of Medicine, St. Louis, MO 63110, USA; 3Instituto de Cancerología Las Américas, Medellín 050025, Colombia; 4Department of Anesthesiology and Surgical Intensive Care Medicine, Medical Faculty Mannheim, Heidelberg University, 68167 Mannheim, Germany; 5Manheim Institute for Innate Immunoscience (MI3), Medical Faculty Mannheim, Heidelberg University, 68167 Mannheim, Germany

**Keywords:** KIR, NK cells, HLA

## Abstract

Natural Killer (NK) cells are innate immune cells that mediate antiviral and antitumor responses. NK cell activation and induction of effector functions are tightly regulated by the integration of activating and inhibitory receptors such as killer immunoglobulin-like receptors (KIR). *KIR* genes are characterized by a high degree of diversity due to presence or absence, gene copy number and allelic polymorphism. The aim of this study was to establish the distribution of *KIR* genes and genotypes, to infer the most common haplotypes in an admixed Colombian population and to compare these *KIR* gene frequencies with some Central and South American populations and worldwide. A total of 161 individuals from Medellin, Colombia were included in the study. Genomic DNA was used for *KIR* and *HLA* genotyping. We analyzed only *KIR* gene-content (presence or absence) based on PCR-SSO. The *KIR* genotype, most common haplotypes and combinations of *KIR* and *HLA* ligands frequencies were estimated according to the presence or absence of *KIR* and *HLA* genes. Dendrograms, principal component (PC) analysis and Heatmap analysis based on genetic distance were constructed to compare *KIR* gene frequencies among Central and South American, worldwide and Amerindian populations. The 16 *KIR* genes analyzed were distributed in 37 different genotypes and the 7 most frequent *KIR* inferred haplotypes. Importantly, we found three new genotypes not previously reported in any other ethnic group. Our genetic distance, PC and Heatmap analysis revealed marked differences in the distribution of *KIR* gene frequencies in the Medellin population compared to worldwide populations. These differences occurred mainly in the activating *KIR* isoforms, which are more frequent in our population, particularly *KIR3DS1*. Finally, we observed unique structural patterns of genotypes, which evidences the potential diversity and variability of this gene family in our population, and the need for exhaustive genetic studies to expand our understanding of the *KIR* gene complex in Colombian populations.

## 1. Introduction

The killer immunoglobulin-like receptors (KIR, also known as CD158) are a highly polymorphic family of transmembrane glycoproteins expressed on the surface of natural killer (NK) cells [1,2] and a few subsets of CD4 and CD8 T cells as well as γδ T cells [3,4,5]. KIR molecules interact with classical human leukocyte antigens class I (HLA-I) molecules, mainly HLA-A, HLA-B and HLA-C; and nonclassical class I molecules, particularly HLA-G and HLA-F [6]. KIR-HLA-I interactions regulate NK cell development and activation and are particularly important to control NK cell education, responsiveness and to ensure self-tolerance. Thus, NK cells lacking the specific inhibitory KIR molecule or missing KIR-HLA-I self-interactions are considered uneducated and hyporesponsive [7]. To compensate for this, a large proportion of KIR negative cells express CD94/NKG2A that binds to HLA-E, and NK cell education and response can be mediated by this interaction [8]. Upon maturation, interaction between inhibitory KIR and HLA-I molecules in mature NK cells prevents NK cell activation. In contrast, downregulation of HLA-I upon viral infections, tumor transformation and transplantation favor NK cell activation and allows them to mediate “missing self” recognition of potential target cells [9,10,11].

The KIR nomenclature is determined according to the structure of the molecules: the number of extracellular Ig-like domains (D) and the length of the intracellular tail whether this is long (L) or short (S). Moreover, based on their ability to activate immunoreceptor tyrosine-based activating motif (ITAM)-containing adapter DAP12 or immune tyrosine-based inhibitory motifs (ITIM)-based signaling pathways, KIR are classified as activating or inhibitory respectively [12].

The genes encoding KIR are located on the chromosome 19q13.4, within a 100–200 Kb region of the Leukocyte Receptor Complex (LRC), one of the most variable regions of the human genome in terms of gene content and sequence polymorphisms [13]. Currently, 13 distinct *KIR* gene loci and two pseudogenes have been described, including *KIR2DL1*, *KIR2DL23*, *KIR2DL4*, *KIR2DL5A*, *KIR2DL5B*, *KIR2DS1*, *KIR2DS2*, *KIR2DS3*, *KIR2DS4*, *KIR2DS5*, *KIR3DL1S1*, *KIR3DL2*, *KIR3DL3*, *KIR2DP1* and *KIR3DP1* [14]. *KIR* genes are classified in two functional haplotypes called A and B, each one showing variation in the number and type of *KIR* genes [15]. Group A or B haplotypes are characterized by the presence of four framework *KIR* genes (*KIR3DL3*, *KIR3DP1*, *KIR2DL4* and *KIR3DL2*) which are present in almost all individuals [16]. Group A haplotypes are generally non-variable in their gene organization, containing the framework genes, inhibitory KIR genes *KIR2DL1*, *KIR2DL3* and *KIR3DL1*, and the activating gene *KIR2DS4*. In contrast, Group B haplotypes are characterized by variations in gene content and they can include one or more of the following genes encoding the activating *KIR2DS1*, *KIR2DS2*, *KIR2DS3*, *KIR2DS5* and *KIR3DS1* and genes encoding the inhibitory *KIR2DL5* and *KIR2DL2* [17]. Based on this, individuals can be assigned to the A/A (homozygous for A), A/B or B/B genotype. Additionally, each haplotype has a region that exhibits a high relative asymmetric recombination rate, which divides the haplotype into a centromeric (Cen) and telomeric (Tel) region. The centromeric segment corresponds to the region including all genes located between *KIR3DL3* and *KIR3DP1* and the telomeric segment consists of all genes present between *KIR2DL4* and *KIR3DL2* [14].

The inhibitory receptors KIR2DL1, KIR2DL2/3, KIR3DL1 and KIR3DL2 are specific for the HLA class I ligands C2, C1, Bw4 and A3/11, respectively. In contrast, KIR2DL4 binds to HLA-G, a non-classical HLA class I molecule. HLA class I ligands for KIR3DL3 and KIR2DL5 are still unknown, nevertheless KIR2DL5 binds to the poliovirus receptor (PVR, CD155) [18,19], and KIR3DL3 binds to HERV–H LTR-associating 2 (HHLA2; also known as B7H5 and B7H7) [20]. Ligands for activating KIR were elusive for many years due to the difficulty to demonstrate KIR/ligand interactions. It is now known that KIR2DS1, KIR2DS2, KIR2DS4 and KIR2DS5 bind specific group 1 and 2 HLA-C molecules, although with lower affinity than the inhibitory KIR. KIR2DS2 and KIR2DS4 also bind HLA-A*11 while KIR3DS1 recognizes HLA-F or HLA-B*51 molecule [6,21].

Combinations of KIR and HLA polymorphisms have been associated with infections, autoimmune diseases, cancers, and pregnancy disorders. For instance, interaction between KIR3DS1 and HLA-Bw4-I-80 isoform is associated with delayed progression to AIDS and HIV-infection [22]; homozygosity of HLA-C1 and KIR2DL3 is associated with resolution of HCV infection [23]; combination of KIR2DS1 and/or KIR2DS2 with homozygosity of HLA-C1 or -C2 favors susceptibility to psoriatic arthritis [24]; KIR2DS2 combined with HLA-C1 in the absence of HLA-C2 and HLA-Bw4 is associated with increased susceptibility to type I diabetes [25]; HLA–KIR gene combinations that seem to favor NK cell activation are associated with predisposition to human papilloma virus (HPV)-induced cervical cancer [26]; whereas HLA–KIR gene combinations that seem to favor NK cell inhibition have been associated with preeclampsia [27]. Furthermore, expression of specific KIR alleles has been associated with better clinical outcomes in patients receiving hematopoietic cells transplant [28,29]. These findings highlight the contribution of HLA-KIR interaction to the outcome of various diseases and support the importance of including KIR analysis in future cell-based immunotherapies [30,31].

A comprehensive analysis of population-specific genes, genotype and haplotype frequencies of *KIR* might help to better understand their evolution and role in immunity. Although *KIR* gene and haplotype profiles have been studied worldwide, *KIR* diversity has not been extensively characterized in Colombia, a country with a high genetic admixture and combinations of genetic ancestry. At the time of sampling in 2019, *KIR* genes haplotypes have been only explored in two Colombian populations with very few individuals [32,33]. These studies analyzed different ancestral groups, with different genetic background. Hollenbach et al. investigated patterns of variation in the *KIR* cluster in 52 populations in the Human Genome Diversity Project—Centre d’Etude du Polymorphisme Human (HGDP-CEPH). They included 110 individuals of 5 Amerindian populations (12 of Colombian Indigenous Piapoco-Curripako). The Amerindian populations did not exhibit high levels of genotypic diversity, but they displayed distinctive patterns unique to the region [32]. Bonilla et al. analyzed *KIR* genotypes in 119 unrelated individuals of Caucasoid and Afro-descendant populations from the Andean and Pacific region of Colombia, and found 59 different profiles [33].

Interest in the characterization of worldwide KIR diversity is ongoing due to the genetic variation of *KIR* being the single most important factor that shapes NK cell function. The available data related to KIR in human populations have increased and the discoveries about *KIR* polymorphism, *KIR* gene content, and copy number/*KIR* allelic variation have guided the research into the impact of *KIR* genetic variation on human health. In Colombian populations, this research has been almost null. The present study aimed to use current knowledge of *KIR* gene profiles to characterize the distribution of *KIR* genes, genotypes, and inferred haplotypes in an admixed Colombian population. Furthermore, we performed an extensive comparison of the *KIR* gene frequencies and genotypes of our population with other populations in our region and worldwide.

Despite being a descriptive study, our results represent one of the first approaches in the genetic variation of KIR research in Colombia, and we encourage future studies to obtain data using next generation technologies, with a greater number of individuals with different genetic background and from different country regions. Expanding our understanding of *KIR* genetic variation in Colombian populations and their comparison with other regions of the world might be useful for future research on ethnicity-based diseases. Furthermore, the understanding of KIR diversity can allow the identification of *KIR* and *HLA* genotypes that influence susceptibility to infectious, autoimmune diseases and the outcome of transplantation in our region.

## 2. Materials and Methods

### 2.1. Population and Samples

A total of 161 unrelated healthy individuals from Medellin, Colombia were included in the study. Medellin is located in the central region of the Andes Mountains in South America and its population shows a wide admixture contribution from ancestral source populations including European (75%), Native American (18%) and African (7%) descents [34].

We carried out a convenience sampling and we included 161 hematopoietic stem cell donors registered in the Instituto de Cancerologia of the Clinica Las Americas, and the Laboratorio de Trasplantes of the Universidad de Antioquia, between 2004 and 2018. The convenience sampling strategy can be used in situations where time is a constraint, because this method allows quick data collection. It also allows researchers to generate more samples with less or no investment and in a brief period. Another characteristic is that elements are easily accessible and samples are available to the researchers.

Between 2004 and 2018, the Instituto de Cancerologia performed 347 transplants, of which 195 attended the Laboratorio de Trasplantes of the Universidad de Antioquia, to carry out the immunological studies. Of these, 161 donor-recipient pairs had DNA samples in optimal conditions for *KIR* typing. As part of the pre-transplant studies, donor-recipient pairs were typed for *HLA-A, -B, -DRβ, -DQβ* and *-DQα* alleles by PCR-SSP (sequence-specific primers) or PCR-SSO (sequence-specific oligonucleotides). In the particular case of the *HLA-C* allele, we had performed genotyping only since 2013 by PCR-SSO (88 pairs had not previously been genotyped for the *HLA-C* allele).

The study protocol was reviewed and approved by the local research Ethics Committee of Universidad de Antioquia and Clinica Las Americas, and all the samples used in this study were collected after an informed consent was obtained. Genomic DNA was extracted from peripheral blood using a modified salting-out procedure as previously reported [35]. DNA was spectrometrically quantified (NanoDrop™ 2000) and the integrity was assessed in a 2% agarose gel with GelRed Nucleic Acid Gel Stain (Biotium, Fremont, CA, USA).

### 2.2. KIR and HLA Class I Genotyping

Genomic DNA was used for *KIR* and *HLA* genotyping. For *KIR* genotyping, the presence or absence of the genes *KIR2DL1*-5, *KIR2DS1*-5, *KIR3DL1*-3, *KIR3DS1*, *KIR2DP1* and *KIR3DP1* was determined based on the oligonucleotide probe-hybridization method using the LIFECODES *KIR*-SSO typing Kit (Immucor, Peachtree Corners, GA, USA). Approximately 50 ng of genomic DNA were used for each PCR reaction and the amplification steps were performed according to the manufacturer instructions. To ensure accuracy in PCR reactions and hybridizations, two different internal controls (designated CON100 and CON200) were used to hybridize *KIR3DP1* and *KIR3DL3* respectively.

For the *HLA* genotyping, the *HLA*-*A* and -*B* genes were assessed using CTS-PCR-SSP Tray Kit (low resolution) or LIFECODES *HLA*-SSO Typing Kits (intermediate-resolution). For *HLA*-C typing we used SSO methodology. Approximately 100 ng of genomic DNA were used for each PCR reaction. Each kit contained two consensus SSO probes (designated CON200 and CON300) that hybridized to all alleles and acted as internal controls to verify amplification and to confirm that hybridizations occurred. PCR products from both kits were hybridized and measured separately using the Luminex 200™ System. For each *KIR*- and *HLA*-SSO typing at least 60 events for each of the Luminex Microspheres were collected.

### 2.3. KIR Gene, Genotype and Haplotype Frequencies Analysis

*KIR* gene frequencies were estimated from carrier frequencies obtained by direct counting. We used the Bernstein’s equation *GF= 1-√(1-CF)*, where CF indicates carrier frequencies. For the same data, the confidence interval (IC) was calculated using the formula *GF ± ε√(GF(1-GF)/n)*, where ε corresponds to standard error [36].

*KIR* genotype frequencies were determined with direct count according to the presence or absence of *KIR* genes. Genotype IDs for each sample were obtained from the Allele Frequency Net Database (AFND) (http://www.allelefrequencies.net, accessed on 28 July 2022). Genotypes that were not identified in the database were reported here as *NPR* (not previously reported). *KIR* haplotype diversity and frequencies were inferred using HAPLO-IHP (Haplotype inference using identified haplotype patterns) software [37]. For the inference of *KIR* haplotypes, we considered 23 reference haplotypes and patterns of linkage disequilibrium between pairs of *KIR* genes (Appendix A) [38]. Several assumptions were made for the analysis: (I) framework genes are always present in each haplotype, (II) if *KIR2DL5* is present then *KIR2DS3* and/or *KIR2DS5* are also present, (III) if *KIR2DL2* is present, *KIR2DL3* is absent and vice versa, (IV) if *KIR2DS4* is present, *KIR3DL1* is also present [39]. We used a set of haplotype patterns labeled with the nomenclature described by Vierra-Green et al. [40]. For instance, ‘cA01~tB01 2DS5’ is a full-length haplotype comprised of the first centromeric A region in *cis* with the first telomeric B region, and this haplotype has the *KIR2DS5* gene [41]. We were not able to confidently identify the less common haplotypes, therefore, for the analyzes we only took into account those haplotypes inferred in at least eight individuals. 

### 2.4. HLA–KIR Combinations in Medellin Population

The frequencies of different combinations of *KIR* and *HLA* ligands (*KIR2DL1* and *HLA*-*C2*, *KIR2DL2*/3 and *HLA*-*C1* [42], *KIR3DL1* and *HLA*-*Bw4* [43], *KIR3DL2* and *HLA*-*A3*/*11* [44,45], *KIR2DS1* and *HLA*-*C2*, *KIR2DS2* and *HLA*-*C1* [42] or *HLA*-*A11*, *KIR3DS1* and *HLA*-*Bw4-80I* [6] and *KIR2DS4F* and *HLA*-*A3*/*11* [46]) were determined by presence and absence of *KIR* and *HLA* genes.

### 2.5. Statistical Analysis

*KIR* carrier frequencies of 41 world populations were obtained from AFND and summarized in Appendix A. We selected the populations if the 16 *KIR* genes were evaluated in at least 100 individuals. Additionally, we included 14 Amerindian populations, and two Brazilian populations with different ancestry, even though they had not studied the frequency of *KIR2DP1* and *KIR3DP1* genes (Appendix A). Genetic distances were calculated by the Nei’s method using PHYLIP software version 3.698 [47]. Distances among Central and South American and worldwide populations were calculated based on 16 *KIR* genes (*KIR2DL1*, *KIR2DL2*, *KIR2DL3*, *KIR2DL4*, *KIR2DL5*, *KIR2DS1*, *KIR2DS2*, *KIR2DS3*, *KIR2DS4*, *KIR2DS5*, *KIR3DS1*, *KIR3DL1*, *KIR3DL2*, *KIR3DL3*, *KIR2DP1* and *KIR3DP1*) carrier frequencies. For Amerindians, the distances were calculated based on 14 *KIR* genes (without *KIR2DP1* and *KIR3DP1)*. Dendrograms based on genetic distance were constructed by the neighbor-joining (NJ) method, and visualized with MEGA-X software [48]. Principal component analysis (PCA) was performed in R version 4.0.3 using *KIR* carrier frequencies data. In the case of PCA biplots, dots represent PC scores of each population and the arrows represent the loadings of the *KIR* genes frequencies. Package “pheatmap” (https://cran.r-project.org/web/packages/pheatmap/index.html, accessed on 5 June 2022) based on statistical software R version 4.0.3, R Core Team, Vienna, Austria (https://www.r-project.org/, accessed on 5 June 2022) was used to draw a Heatmap containing Medellin and 14 other Central and South American populations or 41 worldwide populations with 16 KIR genes. Additionally, we drew a Heatmap containing Medellin and 57 other worldwide populations, including 14 Amerindian populations, with 14 *KIR* genes. The Heatmap is constructed using Hierarchical Clustering algorithm based on Euclidean distance.

## 3. Results

### 3.1. KIR Genes and Genotypes Frequencies in a Colombian Population

To characterize the diversity of *KIR* genotypes in an admixed South American population of Medellin, Colombia, we evaluated the presence or absence of 16 different *KIR* genes in 161 unrelated healthy individuals. Table 1 shows the frequencies of the tested *KIR* genes and their distribution in 37 different genotypes arranged according to the number of loci. As expected, the *KIR* genes *KIR3DL3*, *KIR3DP1*, *KIR2DL4* and *KIR3DL2* were present in all individuals. The most frequent genes detected in the Colombian population were: *KIR2DP1* (77.7%), *KIR2DL1* (77.7%), *KIR3DL1* (75.1%), *KIR2DS4* (73.9%) and *KIR2DL3* (66.6%). The *KIR2DS4* gene has two versions: one with the full-length and another with a deletion of 22bp in exon 5. In the Medellin population, *KIR2DS4F/KIR2DS4F*, *KIR2DS4F/KIR2DS4D* and *KIR2DS4D/KIR2DS4D* were present in 39.8%, 32.3% and 21.1% of all individuals (Gene Calculated Frequencies: 22.4%, 17.7% and 11.2% respectively).

In Appendix A the deeper color represented higher observed frequency. We found all KIR genes are present in the Medellin population. Strikingly, we observed that the Medellin population had high frequencies for *KIR* genes associated to Group B haplotypes, especially *KIR3DS1*. When Medellin *KIR* genes frequencies werecompared with Central and South American populations, we observed that the Uruguayan population had similar frequencies in genes of Group B (Appendix A). Likewise, in worldwide populations, we observed that Medellin had similarities with Uruguayan, Indian [49] and Bogota populations [33]; where *KIR* activating genes were predominant (except for *KIR2DS3*) (Appendix A). When Amerindian populations were included in the analysis, we observed that Warao [50], Yucpa [51] and Wichi [52] ethnic groups were clustered with the Colombian, Uruguayan and Indian populations [49,53] (Appendix A).

We also characterized different *KIR* genotypes according to the set of *KIR* genes present per individual, which were numbered following known patterns of number and combination of genes (http://www.allelefrequencies.net/kir.asp, accessed on 28 July 2022). Genotypes 1, 2, 3, 4 and 14 were the most predominant in our population (63.34% of all individuals). Interestingly, in this study we found three *KIR* genotypes that have not been previously reported (*NPR*) for any other ethnic group. The first genotype was characterized by the presence of *KIR3DL3*, *KIR2DL5*, *KIR3DL1*, *KIR2DP1*, *KIR2DL1*, *KIR3DP1*, *KIR2DL4*, *KIR2DS4* and *KIR3DL2*; in the absence of *KIR2DS2*, *KIR2DL2*, *KIR2DL3*, *KIR2DS3*, *KIR3DS1*, *KIR2DS5* and *KIR2DS1* genes. The second genotype had *KIR3DL3*, *KIR2DS2*, *KIR2DL2*, *KIR2DL3*, *KIR3DL1*, *KIR3DP1*, *KIR2DL4*, *KIR3DS1*, *KIR2DS4* and *KIR3DL2*; in the absence of *KIR2DL5*, *KIR2DS3*, *KIR2DP1*, *KIR2DL1*, *KIR2DS5* and *KIR2DS1* genes. The third genotype was characterized by the presence of *KIR3DL3*, *KIR2DS2*, *KIR2DL2*, *KIR2DL3*, *KIR2DL5*, *KIR3DL1*, *KIR3DP1*, *KIR2DL4*, *KIR3DS1*, *KIR2DS5*, *KIR2DS1* and *KIR3DL2*; and the absence of *KIR2DS3*, *KIR2DP1*, *KIR2DL1* and *KIR2DS4* (Table 1).

### 3.2. Haplotype Frequencies

Based on *KIR* gene content analysis and using previously well-characterized haplotypes [38,39,40,41], we identified the most common haplotypes in the Medellin population (Table 2). The *KIR* reference haplotypes explained 92.24% of the haplotype variation in the Colombian population [37,41]. The most frequent haplotypes were: cA01~tA01 (41.6%); cA01~tB01 2DS5 (16.5%); cB02~tB01 (10.9%); cB02~tA01 (9.9%); cA01~tB05 (5.9%); cB01~tA01 2DS3 (5%) and cB01~tB01 2DS3 (2.5%) (Table 2). In 135 individuals (83.85%) both haplotypes could be resolved with the 7 inferred haplotypes, while in 24 individuals (14.91%) only one haplotype could be assigned.

### 3.3. KIR-HLA Combinations

We also assessed the frequencies of known *HLA I* alleles identified as ligands of KIR molecules and further identified *KIR*-ligands sets more prevalent in our population. *HLA*-*C1* and *HLA*-*Bw4* alleles were predominant in our population and the frequency was in both cases above 85%. In contrast, the frequencies of *HLA*-*C2* and *HLA*-*A11*/*A3* were 58.39 and 22.36%, respectively (Figure 1).

*KIR*-ligand combinations were defined according to Closa et al. [54] considering the presence of *KIR* genes and their respective ligands described so far. For *KIR2DS4*-A3/A11 we only considered full-length variant. The most frequent KIR-ligand pairs were the inhibitory combinations *KIR3DL1*-*Bw4* (81.4%), *KIR2DL3*-*C1* (75.2%) and *KIR2DL1*-*C2* (55.3%). In contrast, activating *KIR*-ligand pairs were expressed in a lower frequency: *KIR2DS4F–A3 A11* (14.3%), *KIR3DS1*-*Bw4-80I* (26.7%), *KIR2DS2*-*C1* (44.7%) and *KIR2DS1*-*C2* (36%) (Table 3).

All tested individuals had a least one inhibitory KIR–ligand pair. When we analyzed the presence of more than one pair, the most common combination was *KIR2DL2*/3-*C1* + *KIR3DL1*-*Bw4* (36%), followed by *KIR2DL1*-*C2* + *KIR2DL2*/3-*C1* + *KIR3DL1*-*Bw4* (22.4%) and *KIR2DL1*-*C2* + *KIR3DL1*-*Bw4* (9.3%). Finally, 19 individuals (11.8%) had all inhibitory KIR–ligand pairs evaluated (Table 4).

### 3.4. Genetic Distance Comparative Analysis with Regional and Worldwide Populations

Next, we compared the diversity of our admixed population from Medellin, Colombia with other populations from Central and South America, including Bogota, Colombia [28]; populations around the word; and Amerindian populations. For this, we performed genetic distances analysis based on the *KIR* gene carrier frequencies. We analyzed publicly available datasets of 43 populations, including 19 American (*n* = 3569), 14 Asian (*n* = 3751), 8 European (*n* = 2196) and 2 African (*n* = 285) populations. We also included 14 Amerindian populations (*n* = 1099).

*KIR* genes frequencies summarized in Appendix A were used to construct dendrograms based on a neighbor-joining (NJ) algorithm to estimate genetic distances between populations based on their gene frequencies and show these distances in a dendrogram that groups populations presenting similar frequencies. In Central and South American populations, we observed three clusters. The three Mexican populations evaluated clustered together with the Chilean and Venezuelan populations [55,56,57]. In addition, three out of the five Brazilian populations analyzed clustered with the Panamanian population [58,59,60] and the remaining two clustered with the Argentinian population [61]. Interestingly, Uruguayan and Colombian populations did not cluster with other Central and South American populations suggesting differences in the genetic composition of *KIR* genes (Figure 2). Between Bogota and Medellin populations the *KIR* gene frequencies variation was approximately 4.5%, while between Uruguayan and Medellin it was approximately 0.75%.

We also constructed a neighbor-joining (NJ) dendrogram to assess potential genetic relationships of the admixed population from Medellin Colombia with 41 worldwide populations (Figure 3). In the dendrogram, we identified four different clusters. A first cluster encompassed the two African populations (South African and Senegal) [62,63], and three Asian populations (Turkey [64], Iran [65] and Saudi Arabia). The European and American populations were combined in two further different clusters. The second cluster grouped French, Italian, Irish, Mexican, Brazilian, North American and Thai populations [55,58,60,63,66,67,68,69]. The third cluster included Brazilian, Argentine, English, Panamanian, Venezuelan, Spanish, Chilean, North American and Mexican populations [55,56,57,61,67,70]. Finally, the fourth cluster included seven of the 14 Asian populations, mainly Chinese populations [71,72,73,74,75,76]. Interestingly, of the two Colombian populations, the Bogota group showed a KIR composition similar to the Indian populations [49,53]. Surprisingly, the Medellin population clustered alone suggesting a particular *KIR* composition in this population (Figure 3).

Finally, we constructed three NJ dendrograms to assess potential genetic relationships of the admixed population from Medellin Colombia with Amerindian populations (Appendix A). In Appendix A we analyzed genetic distances among Colombian and Amerindian populations, and we identified two different clusters. The first cluster encompassed the Colombian populations with Venezuelan [50,51], Mexican [77] and Argentinean [52] ethnic groups. The second cluster grouped Brazilian and Paraguay Amerindian populations [78]. Interestingly, the Bogota group showed similar *KIR* frequencies to the Yucpa tribe from Venezuela [51], while the Medellin population clustered together Warao and Wichi tribes [50,52]. Appendix A shows genetic distances among admixed Central and South American and Amerindian populations, where we observed that the previous clustering patterns hold. Likewise, in Appendix A we note that Bogota–Yucpa and Medellin–Warao distances remained when genetic distances among worldwide and Amerindian populations were evaluated. As we observed in Figure 3, the Lucknow population from India [49], had similar *KIR* frequencies to the Bogota and Yucpa populations.

We used a different approach to evaluate differences between the populations using Principal Component Analysis (PCA) based on the frequencies of the 16 *KIR* genes. This method converts the *KIR* genes frequencies into a set of values of linearly uncorrelated variables (principal components). On the PCA for Central and South America (Figure 4), Colombian populations clustered separately from the other 13 populations. This distribution is mainly driven by differences in two specific KIR: *KIR3DS1* and *KIR2DL3*. We found that Medellin exhibited the highest frequency of *KIR3DS1* (0.741), and this population was positively associated with the variable *KIR3DS1*. *KIR3DS1* frequency was the major contributor to the variability in the Medellin population in relation to Central and South American populations.

On the other hand, the Bogota population had the lowest *KIR2DL3* frequency (0.31), and this population was associated negatively with the variable *KIR2DL3* (Figure 4 and Appendix A). Unlike the NJ analysis, PCA analysis did not show other particular clustering between the Central and South American populations analyzed.

We next performed PCA analysis with the worldwide populations. Despite the marked differences in the NJ analysis, most of the worldwide populations did not show a clear grouping in the PCA with only twelve showing differential clustering. Seven out of the 14 Asian population clustered together in the right side of the biplot according to their highest *KIR2DL3* frequencies. Two Africans populations grouped in upper center according to the lowest *KIR3DS1* frequencies. Bogota clustered with Indian populations in the lower left side. In contrast, Medellin population showed genetic distance from the other populations evaluated, and this result can be explained by the highest *KIR3DS1* frequency observed in this population (Figure 5). These results are in concordance with the dendrograms previously described.

Finally, we performed PCA analysis with the Amerindian populations (Appendix A) based on the frequencies of the 14 *KIR* genes. As observed in Appendix A, Colombian populations clustered with Venezuelan and Argentinean ethnic groups. This distribution is mainly driven by differences in six KIR: *KIR2DL2* and *KIR2DS2*; and *KIR2DL5*, *KIR2DS5*, *KIR2DS1* and *KIR3DS1*. We found that Medellin exhibited higher frequencies of *KIR3DS1* (0.741), *KIR2DL5* (0.689), *KIR2DS1* (0.596) and *KIR2DS5* (0.571) and these frequencies were similar to Warao and Wichi populations. On the other hand, Bogota population had higher *KIR2DL2* (0.68) and *KIR2DS2* (0.58) frequencies, and these were similar to Yucpa tribe (Appendix A and Appendix A). Appendix A shows PCA among worldwide and Amerindian populations and we observed that clustering remained according to the six *KIR* genes frequencies.

## 4. Discussion

Our study is the first to analyze the distribution of *KIR* genes, genotypes and haplotype combinations in an admixed population from Medellin, Colombia. We identified gene and genotypic variations with unique structural patterns that contributed to marked differences in *KIR* gene frequencies between our population and others from Central and South America, worldwide and Amerindians.

The most common *KIR* genes in the admixed population from Medellin, Colombia included *KIR2DL1*, *KIR3DL1*, *KIR2DS4* and *KIR2DL3*, which correlates with the most common *KIR* genes reported in diverse populations across the world. Other *KIR* genes, namely *KIR2DS2*, *KIR2DL2*, *KIR2DS5*, *KIR2DS1* and *KIR2DL5* were frequent in our population compared to reports from Uruguay, Indian and Amerindian populations and comparable to those observed in Central and South American populations [50,55,56,77,79,80,81,82,83].

In the Medellin population, the *KIR2DS3* gene was found in 10.5% of individuals, and the *KIR2DS1* and *KIR2DS5* genes in 36.5% and 34.5% respectively. Our data are in agreement with the results from Bonilla et al., who reported similar frequencies of the *KIR2DS3*, *KIR2DS1* and *KIR2DS5* genes in individuals from the Andean and Pacific region [33]. However, the frequencies of these *KIR* genes are different from those described in four American populations reported by Hollenbach et al. [32]. In particular, in the Piapoco-Curripako population (the Colombian indigenous group), the *KIR2DS3* gene was completely absent. Gendzekhadze et al. also observed that the *KIR2DS3* gene was not present in the Yucpa population from Venezuela [51]. *KIR2DS5* and *KIR2DS3* genes are located at the same *locus*, which is duplicated in the centromeric and telomeric motifs of the Bx haplotypes. Worldwide, in centromeric motifs, the *KIR2DS3* gene occurs more frequently than the *KIR2DS5* gene, while in Amerindian populations a fixation of the *KIR2DS5* allele was evidenced [32,50]. These differences may be due to genetic isolation of these populations, in addition to the limited number of individuals studied (*n* = 12 and 61 respectively).

Our results of *KIR* gene frequencies show a genetic organization in the Medellin population where the activating *KIR* are dominant compared to the inhibitory *KIR* isoforms (Table 1 and Appendix A). The low frequencies of the *KIR2DL2*, *KIR2DL3*, *KIR2DL1*, *KIR3DL1* and *KIR2DS4* genes are in agreement with the low frequencies of haplotype A in our population (13% haplotype A and 87% of haplotype B). These percentages are similar to other populations in Central and South America, where there is a predominance of Bx haplotypes [50,55,56,77,79,80,81,82,83]. Comparative analyses of *KIR* haplotypes by the Rajalingam group found an association between patterns of human prehistoric migration and the distribution of *KIR* haplotypes in different human populations [84]. Based on the distribution of *KIR* A and B haplotypes, the Rajalingam group divided the populations into three groups: populations carrying dominant A haplotypes (Asian populations), populations carrying dominant B haplotypes (American and Australian populations), and populations carrying balanced A and B haplotypes (European populations). A significant finding to emerge from this investigation is the clear dominance of *KIR* B haplotypes in the Medellin population, which coincides with Rajalingam patterns.

KIR diversity among different populations and ethnicities has been attributed to variations in genotype and haplotype frequencies. Additionally, polymorphisms in *KIR* genes, variations in gene content and gene copy numbers of each *KIR* haplotype also contribute to diversity in KIR expression [85]. There is a set of genotypes and haplotypes that typically explains up to 90% of the variations observed worldwide, even in geographically distant populations [37,86]. However, extensive variation in genotypes and haplotypes as well as virtual absence of the most common genotypes in some populations denotes the plasticity and high degree of variation in the gene content of the human *KIR* cluster [32].

Interestingly, we found three *KIR* genotypes that have not been previously described in the literature (Table 1). Among the new genotypes, *NPR1* had nine *KIR* genes, which is partially similar to the genotype described for Arnheim et al. [87] in a Swedish population. However, in their study, they did not confirm the presence of the *KIR2DP1* and *KIR3DP1* genes. *NPR2* genotype had ten *KIR* genes and was found to be similar to the genotype described in an individual from the Cook Islands [88], but unlike our data, the *KIR2DP1* gene was present. Finally, the gene composition pattern of *NPR3* genotype did not show similarities with any other genotypes described to date. These results support the idea that in Central and South American populations, the genotypic diversity is not high, but there are unique patterns found in the region.

In relation to haplotype frequencies in South American populations, previous works have shown a limited number of *KIR* gene content haplotypes in Amerindians populations (the estimated *KIR* gene-content haplotypes are shown in Appendix A). Taken together, these results validate our inferred haplotypes with the HAPLO-IHP software (the three most common haplotypes in our population were observed in Amerindian populations of Central and South America). However, family segregation studies or long-range sequencing are needed to identify unknown haplotype structures or to precisely infer those haplotypes observed in lower frequencies. For example, de Brito Vargas et al. [78] identified haplotypes carrying large structural deletions or duplications involving multiple *loci.* Moreover, the low-frequency haplotypes are usually variations of the most common ones, differing by only a few alleles in some specific genes (allelic variation on *KIR3DL3* and *KIR3DL1* genes on centromeric and telomeric haplotypes respectively).

In relation to studies on *KIR* haplotypes, two important points are worth noting. First, in Amerindians populations have demonstrated a reduced *KIR* allelic richness compared with worldwide populations [50,51,78]. It must be taken into account that, at the allelic level, there may be low KIR diversity in our population. Second, our study population could exhibit an overall great population differentiation. The admixture and genetic heterogeneity are possibly resulting from different migration processes from other continents (e.g., during the Spanish colonization) and within the country, suggested by the results of our genetic distance analysis. Indeed, Conley et al. [89] used a dataset of 239,989 SNP in a comparative analysis of two Colombian populations (Chocó and Medellin), and they demonstrated that the Medellin population (*n* = 94) showed admixture events, with European–Native American admixture followed by subsequent African admixture. Additionally, the analysis of subcontinental ancestries in the study of Conley et al. [89] revealed that the African ancestry of Medellin was intermediate between Nigerian and other West African populations; the European fraction of Medellin was similar to the Spanish population; and Native America ancestry showed that the Medellin population grouped together with the Colombian Native American populations: Embera, Waunana, Arhuaco, Kogi and Wayuu. These findings underline a similar ancestral source population (in particular Native American Ancestry), with different relative frequencies of African and European ancestors, and these differences could explain variations in *KIR* gene frequencies among Colombian populations [89].

The analysis of several *KIR-HLA* combinations revealed that inhibitory combinations were the most frequent in the Medellin population. All the individuals in the Medellin population had at least one *KIR-HLA* pair, and 83.23% have two or three inhibitory *KIR-HLA* combinations. *KIR3DL1*/*Bw4* pair was present in 131 individuals in our cohort (81.4%). This frequency is above the worldwide frequencies, in contrast to e.g., American cohorts, in which about 29.4% (Veracruz—Mexico) to 68.6% (Havana, Cuba) of the individuals have the *KIR3DL1*/*Bw4* pair (http://www.allelefrequencies.net/KIR-HLA/stats.asp, accessed on 28 July 2022). *KIR3DL1*/*Bw4* (*HLA-A*) and *KIR3DL1*/*Bw4* (*HLA-B*) were present in 48.5% and 53.4% individuals respectively. These frequencies are higher than those observed in Amerindian populations, among which Guarani Ñandeva and Kaingang from Ivaí have the highest frequencies (5% and 13% for *KIR3DL1*/*Bw4* (*HLA-A*) and *KIR3DL1*/*Bw4* (*HLA-B*) respectively) [78]. *KIR2DL3*/*C1* pair was the second most common inhibitory combination (75.2%). This frequency is similar to that observed in worldwide populations, where the lowest values reported are 40% in Xhosa ethnic group from South Africa [90], and 41% in Kaiowá and Mbya Guaraní populations [78], while the highest values reported are 92% and 95.7% in Hong Kong and Singapore populations [91]. *KIR2DL1*/*C2* was observed in 55.3%, similar to the frequencies observed in Central and South American populations (lowest: 39.4% in Curitiba, Brazil individuals [92]; highest: 76.7% in Belo Horizonte, Brazil individuals). In Amerindians, the *KIR2DL1*/*C2* frequencies are lower, ranging from 6% in Guarani Mbya to 26% in Aché from Paraguay [78]. Finally, the *KIR2DL2*/*C1* frequency in our population was 44.7%. In Central and South American populations these frequencies range from 23.5% in Curitiba [92] to 61% in Montevideo, Uruguay (http://www.allelefrequencies.net/KIR-HLA/stats.asp, accessed on 28 July 2022). Furthermore, KIR2DL2 and KIR2DL3 exhibit capability to bind HLA-C C2 allotype [93], although with low affinity. *KIR2DL2*/*C2* and *KIR3DL2*/*C2* frequencies were 29.8% and 50.9% respectively (data not shown).

Our genetic distance and PCA data presented in Figure 2, Figure 3, Figure 4 and Figure 5 confirmed that *KIR* gene frequencies in our South American population are different compared with worldwide populations. Particularly in PCA, the first principal component (PC1) identified strongly correlated with *KIR2DL3* and *KIR2DL2* genes whereas the PC2 strongly correlated with *KIR3DS1* gene. Medellin population showed genetic distance from the other populations in both PCA biplots and these differences may be explained by the highest frequency of *KIR3DS1* gene (71.4%) when compared to the others populations (Appendix A). *KIR3DS1* represents the only activating receptor with three extracellular domains and several studies have identified this gene to be associated with the outcome of various diseases, including viral infection like HIV-1 (slower progression to AIDS and lower viral load in individuals carrying *KIR3DS1* and *HLA*-*B Bw4 I80*) [22], HBV (increased rate of spontaneous recovery) [94] and HCV (viral clearance and sustained virological response) [95]. Moreover, *KIR3DS1* expression is associated with several malignancies including hepatocellular carcinoma, Hodgkin´s lymphoma and respiratory papillomatosis (reduced risk) [96] or cervical neoplasia (increased risk) [26].

Finally, with regard to hematopoietic stem cell transplantation (HSCT), Gabriel et al. [97] showed a decreased progression-free survival of patients carrying *KIR3DS1* with multiple myeloma after autologous transplantation. In contrast, a beneficial effect of *KIR3DS1* in the context of unrelated HSCT has also been observed, showing a decreased acute graft-versus-host disease (GVHD) from transplantation with *KIR3DS1*+ donors [98]. Due to the high frequency of *KIR3DS1* in the Medellin population, it would be important to evaluate its association with the previously listed diseases. In fact, in the context of identical related HSCT, we found that transplant patients whose donors were *KIR2DS1*+ and *KIR3DS1*+ had a 92% and 97% lower risk of acute GVHD respectively (unpublished data).

Using *KIR* as the panel of genetic marker, the results showed that Medellin had a close relationship with the Warao and Wichi ethnic groups, and we found three to five activating *KIR* and one to two inhibitory *KIR* genes of group B haplotype (*KIR2DL5*, *KIR2DS5*, *KIR2DS1* and *KIR3DS1*) were responsible for it. In previous studies, it has been argued that group B haplotypes may be under strong diversifying selection processes, related to several factors including reproduction, unfavorable genes that carry risk of autoimmunity or the creation or loss of novel functional genes to infection responses [86]. These processes ensure that the large majority of individuals within a population have different *KIR* genotypes, a situation analogous to that seen for HLA, and this diversity in human *KIR* genotype is probably the result of natural selection [54,99].

Our study has several limitations. First, we carried out a convenience sampling, so the results could not be generalized to a larger population. The Medellin population is highly mixed and potential bias of the sampling technique exists due to under-representation of subgroups in the sample. Second, genotyping methods used based on SSP and SSOP are prone to errors. HLA-SSP genotyping depends on the quality and quantity of the DNA used, therefore, PCR amplification failure could lead to a failure in typification result. Meanwhile, SSO genotyping depends on the hybridization process, and therefore, low bead count or low median fluorescent intensity value can lead to typing misunderstanding. Additionally, these methods cannot identify new alleles and allelic diversity, so there may be a deviation in the results on the genotypic and haplotypic diversity. Third, for NJ and PCA analyses, we included studies with heterogeneous methodologies and these may contain inaccurate data. Moreover, internal nodes of a dendrogram based on genetic distances do not necessarily represent common ancestry, and the similarities and differences between populations can be consequences of demographic factors. Despite these limitations this represents the first study to examine KIR diversity in an admixed Medellin—Colombian population. However, these findings will require validation in larger, multicenter datasets, with stratified random sampling and high-resolution genotyping technologies.

## 5. Conclusions

We report, for the first time, an analysis of *KIR* genotypes; observed carrier frequency; estimated genetic frequency and inferred haplotype patterns in an admixed Medellin—Colombian population. In particular, estimation of haplotype frequencies and KIR-HLA combinations as well as genetic distance and principal component analysis (PCA) with several population groups from Central and South America, worldwide and Amerindian populations revealed distinct patterns of *KIR* genes and haplotypes in this population. Currently, efforts have been made to understand NK cell function and diversity through genetic studies and their association with disease susceptibility. Our findings on *KIR* genetic diversity and frequencies of *KIR-HLA* pairs in the Medellin population are likely to shed light on the formation of KIR repertoires and the potential implications in processes of NK cell education and function, and their impact in immune response against viral infections, anti-tumor response or transplant outcomes, however these associations remain unexplored in Colombia.

## Figures and Tables

**Figure 1 cells-11-02776-f001:**
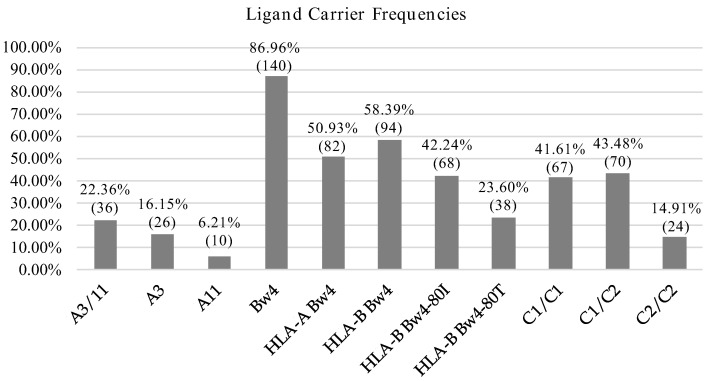
HLA allotype frequencies. Percentage and number of individuals carrying KIR ligands.

**Figure 2 cells-11-02776-f002:**
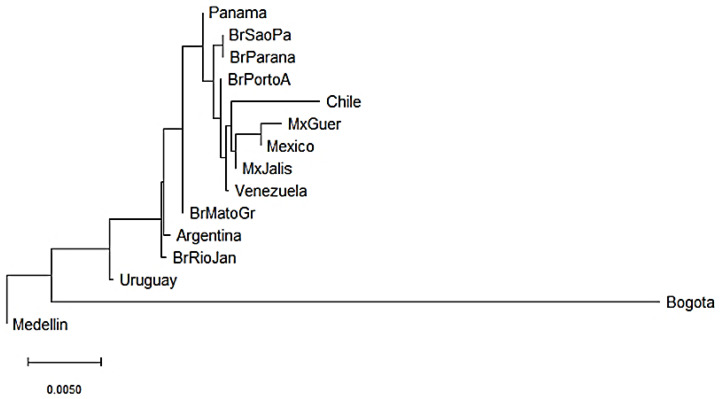
Neighbor-joining (NJ) dendrogram of Nei’s genetic distances among Central and South American populations, based on the KIR genes frequencies. The NJ dendrogram was constructed with 15 American populations from Brazil (*n* = 1327), Mexico (*n* = 544), Argentina (*n* = 365), Venezuela (*n* = 205), Panama (*n* = 116), Chile (*n* = 90), Uruguay (*n* = 41), Bogota—Colombia (*n* = 119) and Medellin—Colombia (*n* = 161). The bar suggests a 0.005 (0.5%) *KIR* gene frequencies variation for the length of the scale.

**Figure 3 cells-11-02776-f003:**
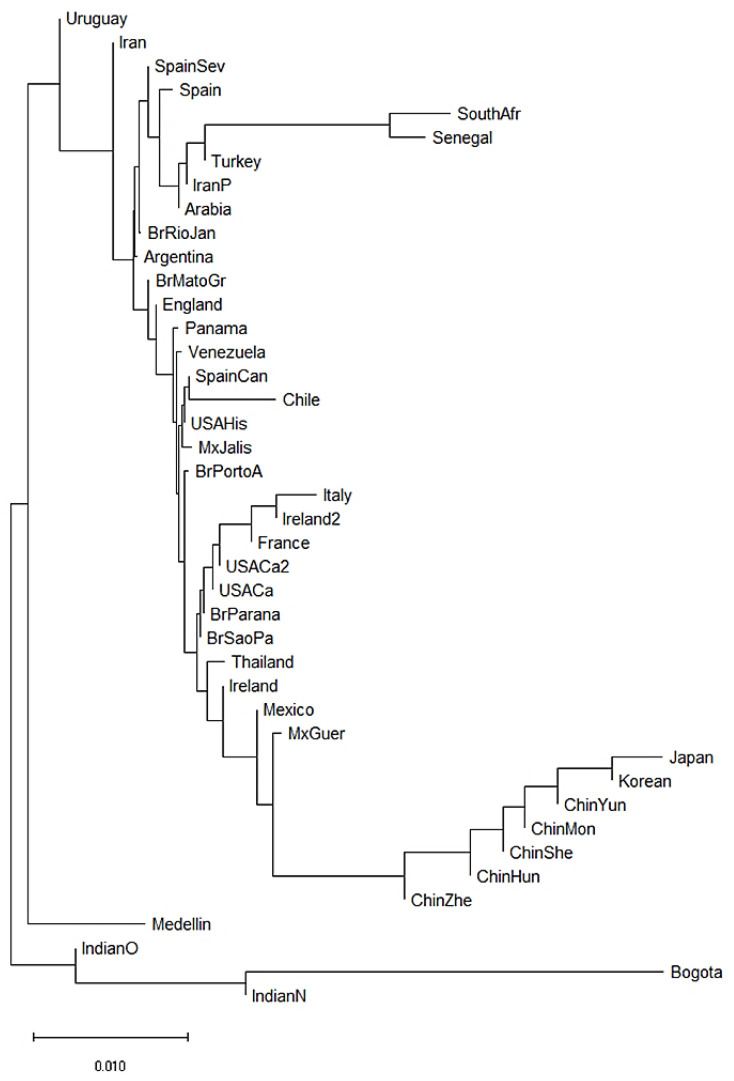
Neighbor-joining (NJ) dendrogram of Nei´s genetic distances among worldwide populations, based on the KIR genes frequencies. The NJ dendrogram was constructed with 42 populations from China (*n* = 1788), Brazil (*n* = 1327), Spain (*n* = 847), India (*n* = 673), England (*n* = 584), United States (*n* = 578), Mexico (*n* = 544), Iran (*n* = 448), Ireland (*n* = 440), Argentina (*n* = 365), Thailand (*n* = 235), Italy (*n* = 237), Venezuela (*n* = 205), South Africa (*n* = 167), Arabia (*n* = 162), South Korea (*n* = 159), Turkey (*n* = 154), Japan (*n* = 132), Senegal (*n* = 118), Panama (*n* = 116), France (*n* = 108), Chile (*n* = 90), Uruguay (*n* = 41), Bogota—Colombia (*n* = 119) and Medellin—Colombia (*n* = 161). The bar suggests a 0.01 (10%) *KIR* gene frequencies variation for the length of the scale.

**Figure 4 cells-11-02776-f004:**
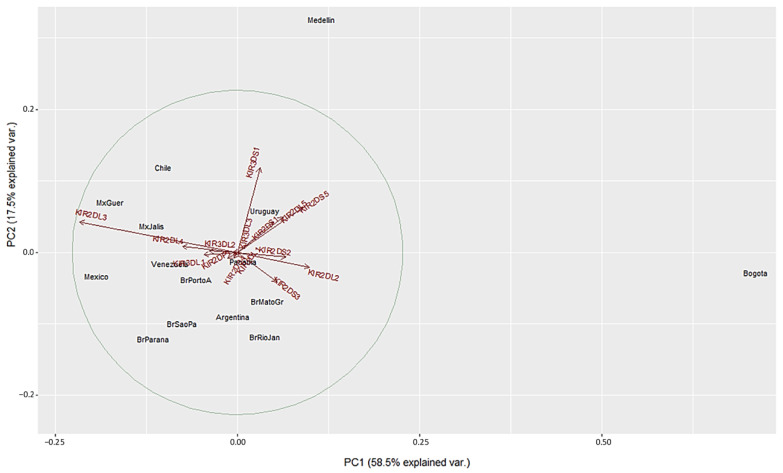
Principal component analysis based on 16 *KIR* genes for Central and South American populations. The circle indicates the correlation between *KIR* gene frequencies. Red arrows show the relative contribution of each *KIR* gene frequency to the variability along the first two axes (PC1 and PC2). PC1 was strongly correlated with *KIR2DL3* and *KIR2DL2* genes and PC2 was strongly correlated with *KIR3DS1* gene.

**Figure 5 cells-11-02776-f005:**
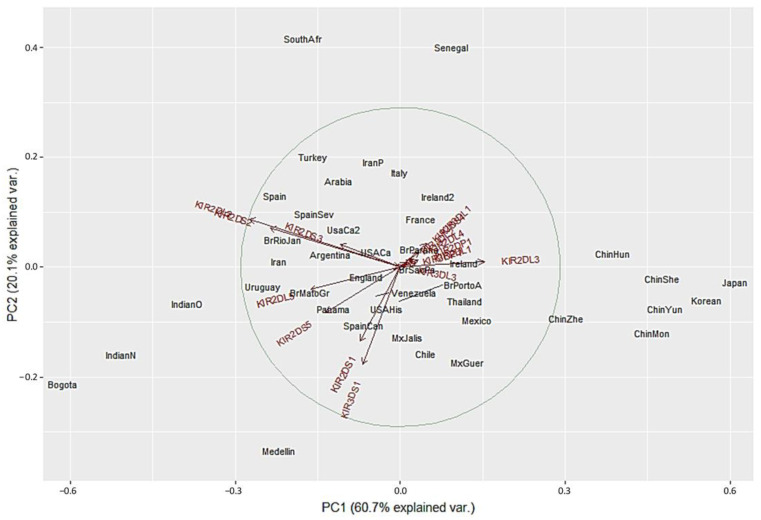
Principal component analysis for worldwide populations. The circle indicates the correlation between *KIR* gene frequencies. Red arrows show the relative contribution of each *KIR* gene frequency to the variability along the first two axes (PC1 and PC2). PC1 was strongly correlated with *KIR2DL3* and *KIR2DL2* genes and PC2 was strongly correlated with *KIR3DS1* gene.

**Table 1 cells-11-02776-t001:** *KIR* gene and genotype frequencies.

		A Haplotype Member				B Haplotype-Specific							Framework/Pseudogenes							
Genotype	Genotype ID	2DL1	2DL3	3DL1	2DS4	2DL2	2DL5	2DS1	2DS2	2DS3	2DS5	3DS1	2DL4	3DL2	3DL3	2DP1	3DP1	N° *loci*	*n*	Genotype Frequencies (%)
BX	2																	13	31	19.25
BX	3																	15	28	17.39
AA	1																	9	20	12.42
BX	14																	10	12	7.453
BX	4																	11	11	6.832
BX	6																	16	8	4.969
BX	71																	12	4	2.484
BX	5																	13	4	2.484
BX	68																	13	4	2.484
BX	7																	15	4	2.484
BX	69																	11	3	1.863
BX	76																	12	3	1.863
BX	72																	8	2	1.242
BX	188																	12	2	1.242
BX	13																	14	2	1.242
BX	70																	14	2	1.242
BX	*NPR1* ^1^																	9	1	0.621
BX	*NPR2* ^1^																	10	1	0.621
BX	106																	11	1	0.621
BX	30																	11	1	0.621
BX	16																	11	1	0.621
BX	43																	11	1	0.621
BX	228																	12	1	0.621
BX	394																	12	1	0.621
BX	27																	12	1	0.621
BX	*NPR3* ^1^																	12	1	0.621
BX	81																	13	1	0.621
BX	8																	13	1	0.621
BX	680																	13	1	0.621
BX	18																	14	1	0.621
BX	12																	14	1	0.621
BX	9																	14	1	0.621
BX	118																	14	1	0.621
BX	90																	14	1	0.621
BX	401																	14	1	0.621
BX	73																	15	1	0.621
BX	58																	15	1	0.621
Direct Count		153	143	151	150	87	111	96	86	32	92	115	161	161	161	153	161			
Carrier Frequencies (%)		95	88.8	93.8	93.2	54	68.9	59.6	53.4	19.9	57.1	71.4	100	100	100	95	100			
Gene Frequencies (%)		77.7	66.6	75.1	73.9	32.2	44.3	36.5	31.8	10.5	34.5	46.5	100	100	100	77.7	100			
Confidence Interval (%)		73.2–82.2	61.4–71.7	70.4–79.8	69.1–78.7	27.1–37.3	38.8–49.7	31.2–41.7	26.7–36.8	7.1–13.8	29.3–39.7	41.1–52	100	100	100	73.2–82.2	100			

^1^ *NPR*: Not previously reported.

**Table 2 cells-11-02776-t002:** Most frequent KIR inferred haplotypes.

		Centromere							Telomere									Sample (2*n* = 322)	
Haplotype	Nomenclature	3DL3	2DS2	2DL2	2DL3	2DP1	2DL1	3DP1	2DL4	3DL1	3DS1	2DL5	2DS3	2DS5	2DS4	2DS1	3DL2	*n*	Frequency (%)
A1	cA01~tA01																	134	41.615
B1	cA01~tB01 2DS5																	53	16.46
B2	cB02~tB01																	35	10.87
B3	cB02~tA01																	32	9.938
B4	cA01~tB05																	19	5.901
B5	cB01~tA01 2DS3																	16	4.969
B6	cB01~tB01 2DS3																	8	2.484

**Table 3 cells-11-02776-t003:** KIR –HLA combinations.

*KIR*/Ligand		Present/Present *n* (%)	Present/Absent *n* (%)	Absent/Present *n* (%)	Absent/Absent *n* (%)
**Inhibition**	2DL1/C2	89 (55.279)	64 (39.752)	5 (3.106)	3 (1.863)
	2DL2/C1	72 (44.720)	15 (9.317)	65 (40.373)	9 (5.590)
	2DL3/C1	121 (75.155)	22 (13.665)	16 (9.938)	2 (1.242)
	3DL1/Bw4	131 (81.367)	19 (11.801)	9 (5.590)	2 (1.242)
	3DL1/Bw4 (HLA-B)	86 (53.416)	64 (39.752)	8 (4.969)	3 (1.863)
	3DL1/Bw4 (HLA-A)	78 (48.447)	72 (44.720)	4 (2.485)	7 (4.348)
	3DL2/A3 A11	35 (21.739)	126 (78.261)	0 (0.00)	0 (0.00)
	3DL2/A3	26 (16.149)	135 (83.851)	0 (0.00)	0 (0.00)
	3DL2/A11	10 (6.211)	151 (93.789)	0 (0.00)	0 (0.00)
**Activation**	2DS1/C2	58 (36.025)	38 (23.603)	36 (22.360)	29 (18.012)
	2DS2/C1	72 (44.720)	14 (8.696)	65 (40.373)	10 (6.211)
	2DS2/HLA-C*16	5 (3.106)	81 (50.310)	8 (4.969)	67 (41.615)
	2DS2/A11	3 (1.863)	83 (51.553)	7 (4.348)	68 (42.236)
	2DS4F/A3 A11	23 (14.286)	93 (57.764)	13 (8.074)	32 (19.876)

**Table 4 cells-11-02776-t004:** Inhibitory *KIR* and *HLA* ligand pairs.

	*KIR* + *HLA* Ligand Pairs	Number Individuals (%)	
1	*KIR2DL1 + C2*	2 (1.242)	8 (4.969)
	*KIR2DL2/3 + C1*	4 (2.485)	
	*KIR3DL1 + Bw4*	1 (0.621)	
	*KIR3DL2 + A3/11*	1 (0.621)	
2	*KIR2DL1 + C2* *KIR2DL2/3 + C1*	10 (6.211)	88 (54.658)
	*KIR2DL1 + C2* *KIR3DL1 + Bw4*	15 (9.317)	
	*KIR2DL1 + C2* *KIR3DL2 + A3/11*	1 (0.621)	
	*KIR2DL2/3 + C1* *KIR3DL1 + Bw4*	58 (36.025)	
	*KIR2DL2/3 + C1* *KIR3DL2 + A3/11*	3 (1.863)	
	*KIR3DL1 + Bw4* *KIR3DL2 + A3/11*	1 (0.621)	
3	*KIR2DL1 + C2* *KIR2DL2/3 + C1* *KIR3DL1 + Bw4*	36 (22.360)	46 (28.572)
	*KIR2DL1 + C2* *KIR2DL2/3 + C1* *KIR3DL2 + A3/11*	2 (1.242)	
	*KIR2DL2/3 + C1* *KIR3DL1 + Bw4* *KIR3DL2 + A3/11*	4 (2.485)	
	*KIR2DL1 + C2* *KIR3DL1 + Bw4* *KIR3DL2 + A3/11*	4 (2.485)	
4	*KIR2DL1 + C2* *KIR2DL2/3 + C1* *KIR3DL1 + Bw4* *KIR3DL2 + A3/11*	19 (11.801)	19 (11.801)

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
