# Peer review of "Killer-Cell Immunoglobulin-like Receptor Diversity in an Admixed South American Population"

_cells, 2022, doi:10.3390/cells11182776_

Round 1
Reviewer 1 Report (Previous Reviewer 2)
Opinion on revised manuscript 1871761 „Killer-Cell Immunoglobulin-Like Receptor Diversity in an Admixed South American Population” by Castrillon et al.
The Authors corected their manuscript remarkably. However, new errors appeared:
1. In the Introduction (130-131) they write: „our results represent one of the first approaches in the genetic variation of KIR research” which is not true – such studies have been published for decades since KIR system was discovered. The Authors should specify that they mean a comprehensive study of KIR gene, genotype and haplotype frequncies as well as their combinations with HLA ligands, and comparison of their population with all aother populations described so far.
2. Results (292-293): „the most common combination was KIR2DL2/3-C1 + KIR3DL1-Bw4 (33.5%)” – however, Table 4 shows 36.02% for this combination. Similarly, KIR2DL1-C2 + KIR3DL1-Bw4 (10,5%) in the text, but 9.3% in Table 4. „Finally, 14 individuals (8.7%) had all inhibitory KIR-ligand pairs evaluated (Table 4)” but Table 4 shows 19 such individuals (11.8%).
3. Discussion (454-456): „…the low frequencies of haplotype A in our population (13% haplotype A and 87% of haplotype B). These expression profiles are similar to other populations…” These are percentages, not expression profiles: Gene expression means mRNA or protein production.
4. Discussion (537-539, 546-547): „These frequencies are higher than observed in Amerindian populations, among which Brazilian individuals of Japanese ancestry have the highest frequencies” – Brazilians with Japanese ancestry have nothing in common with Amerindians except for ancient migration 15 000 years ago of Amerindian ancestors from Asia, from which Japanese also originate.
5. Discussion (575-586); „In previous studies, it has been argued 578 Cells 2022, 11, x FOR PEER REVIEW 17 of 22 that group B haplotypes may be under strong diversifying selection processes, related to 579 several factor including reproduction, unfavorable genes that carry risk of autoimmunity, 580 or to creation or loss of novel functional genes to infection responses” etc.: In this context, you may mention the rapid evolution (i.e., appearance of new HLA-B alleles) in South American Amerindians (see Lienert and Parham, Evolution of MHC class I genes in higher primates. Immunol Cell Biol . 1996 Aug;74(4):349-56. doi: 10.1038/icb.1996.62). As KIR3DL1 interacts with HLA-B(Bw4), this would be a relevant information.
In conclusion, although the manuscript has been corrected, it needs a minor revision again.
Author Response
Please see the attachment

Reviewer 2 Report (New Reviewer)
Killer-Cell Immunoglobulin-Like Receptor Diversity in an Ad-2 mixed South American Population
Marlon Castrillon et al.
Cells-187-1761
This is the study of KIR gene content diversity in a population sample from Medellin in Columbia. KIR modulate functions of natural killer and T cells of immunity through interaction with polymorphic HLA class I. Assessing KIR genotypes and population diversity is important for several diseases and for some types of transplantation matching. The study approach and methods are very outdated, but nevertheless provide potentially useful data for future work. The previous reviewers very thoroughly and knowledgeably identified problems with the study, which have been addressed to some extent (ie less thoroughly) by the authors; thus in essence I would say its not there yet.
Major points
There are multiple places where the corrections claimed in the response to reviewers did not actually make it into the text, or if so then not consistently (eg Number of KIR genes, ligand for KIR3DS1*, accurate referencing etc).
(*the ligand is HLA-F: Garcia Beltran et al 2016 - PMID: 27455421)
The point about sampling was not addressed well: L46: ‘We carried out a convenience sampling and we included 161 individuals.’ I have no idea what this means. The sampling strategy is an important point that needs clarification.
In addition to the previous comments, I have reservations concerning the analysis done. In addition to the technical concerns raised, there are multiple problems with making trees of populations from single locus data (ie the KIR locus), particularly for a locus under selection. That the study population is admixed (as are many others in the data set used) adds further confusion to an already murky picture. Similarly, the PCA analyses are done using non-independent data (the individual KIR genes that make up the KIR locus displaying strong LD). All these reservations are evident in the unusual clusterings of Figure 3 and impossible to interpret Figure 5. I would suggest these two figures go as Supplementary if at all.
The discussion is very long. The simple take home message that the observed KIR gene frequencies/haplotypes are consistent with the origins of the groups that admixed to make up the present Medellin population is somewhat lost in all that.
Minor points:
The order that the genes are shown in in table 1 is baffling: it is neither alphanumerical or gene order (I think it is supposed to be gene order, but 3DL1 is out of place?)
Some KIR2DL2 also bind to C2 in addition to C1 (Hilton 2015, ref 37), so this should be factored into Table 3 and 4 or at least mentioned in discussion. Also, similarly to 2DS4 there are common alleles of other KIR that don’t function as receptors or have reduced function.
L49: “Thus, NK cells lacking of specific inhibitory KIR molecule or missing KIR- HLA-I self-interactions are considered uneducated and hyporesponsive [7].” This is not quite true: a large proportion of KIR negative cells are compensated by CD94/NKG2A interaction with HLA-E (PMID: 26284479)
“Ligands for KIR3DL3 and KIR2DL5 are still unknown and remain to be identified [6].”
Also not true: PMID: 31308249, PMID: 35749424 and PMID: 33536266
L86: “It is now known that KIR2DS1, KIR2DS2, KIR2DS3, KIR2DS4 and KIR2DS5 bind specific group 1 87 and 2 HLA-C molecules, although with lower affinity than the inhibitory KIR.”
By contrast, I’m not aware of the KIR2DS3 ligand (happy to be corrected)
L444: “Gendzekhadze et al. also observed that KIR2DS3 gene was retained in Yucpa population from Venezuela [46]. “
This is not correct; they showed it was absent.
Supplementary Table 3. “Amerindians KIR gene frequencies:”
I doubt that the Brazilian Europeans and Japanese are considered as Amerindian?
Author Response
Please see the attachment

This manuscript is a resubmission of an earlier submission. The following is a list of the peer review reports and author responses from that submission.
Round 1
Reviewer 1 Report
This study describes the KIR presence and absence variation of a population sample from Colombia. Although this reviewer sees value in descriptive population studies like this, several points reduce the significance of this manuscript. On the positive side, this population has not been described before, and the authors have done a fair exploration of the data. However, the study is based on outdated genotyping methods that provide the lowest resolution, limiting discussion and impact. In addition, the method for haplotype inference is highly inaccurate for this type of data, and there were no apparent criteria for choosing the populations used for comparisons. It is difficult to identify a direction in the manuscript, except for the heavy description of the data and re-description of the published data. Throughout the entire manuscript, there are numerous issues, including the approach, analysis, terminology, and references. Please see below the major issues raised during the review.
Abstract
- I suggest avoiding using the term “Latin American” for population genetics studies as Latin America does not reflect ancestry.
- Please clearly state in the abstract that this study analyses only gene-content (presence and absence) based on PCR-SSOP.
- “Further, our data showed that in our population, activating KIR isoforms are more prominent.” This sentence is vague. More prominent compared to other populations or within this population? Is there a statistical test to show that these increased frequencies are significant? What is the point of this statement?
- There is no point in affirming that all genes were found in this population. This same observation will be for the vast majority of the worldwide populations.
- Similarly, the last sentence/conclusion is vague.
Introduction
- The KIR official nomenclature does not include “s” (“receptors”; “KIRs”)
- “KIR receptor” is redundant, as the last R of the acronym is also “receptor”. KIR molecule, for example, would be more appropriate.
- Several references are highly inappropriate. A few examples:
- References 1 and 2 are not the original citations for characterization of KIR
- Reference 6 does not cover the ligands cited in this statement. Several studies discovered these interactions. Hansasuta et al. only presented the interaction of KIR3DL2 with HLA-A3 and HLA-A11.
- The haplotypes A and B were initially proposed by Uhrberg et al., 1997 (PMID: 9430221)
- Line 86. No citations for any of these diseases?
I suggest carefully reviewing the citations to ensure each statement is precise and the original authors are appropriately acknowledged.
- “The structure and nomenclature of the KIRs family is determined according to three factors”. I suggest rewriting: “The KIR nomenclature is determined according to the structure of the molecules”. In addition, sequence similarity was not one of the factors for KIR nomenclature. At that time, there was minimal sequence information, and the order of the gene names was according to the order they were discovered (e.g., KIR2DL1, KIR2DL2, etc.).
- Although the number of KIR genes is confusing and without a clear consensus in the literature, it has been accepted that there are 13 KIR genes and two pseudogenes. KIR3DL1S1 and KIR2DL23 (no slashes) are each ones a single gene.
- Line 64. The parenthesis “(AA and B/x)” is incorrect here. A/A and B/x (slash required for both) are genotypes, not haplotypes.
- Line 68. gen = gene. Haplotypes A are not completely fixed. Some of them miss one or a couple of genes. It is better to list the genes and use caution to affirm they are “fixed”.
- Line 72 – “Based on this, individuals can be assigned to the A/A genotype (homozygous for A) or B/x (x can be either an A or B haplotype).” This limitation of only distinguishing A/A from B/x is due to technical limitations from works using outdated methods for genotyping only the presence and absence of KIR. Recent methods can identify haplotypes.
- Lines 87-89 state that studying KIR variation might help understand evolution and immunity. The following sentence says that KIR are important for transplantation. Transplantation is a recent and artificial therapy that has no effect on KIR evolution and their role in immunity. Several better relevant examples could be used, including KIR in reproduction, diseases (particularly infection), and functional mechanisms
- Line 95. What do the authors mean by “genetic heritage”?
- Introduction lacks information to justify this study and place it within the context of previous work. For example, the authors state that only two studies have described KIR in Colombia. Did these studies analyze the same ancestral groups? Are there differences between the genetic background of these samples? Different locations? Did previous studies find high KIR diversity? In addition: why is it important to characterize the Colombian (or any) population? How does a descriptive study like this contribute to advancing the system's understanding?
Methods/Results
- The authors describe that the Medellin population is stratified, as most of the highly mixed populations in South America. How did the authors address the sampling bias? Were there inclusion/exclusion criteria? If the authors did not analyze a single stratum, how do we know that, for example, the representation of majoritarian African or Native Americans was not artificially inflated?
- Please clarify the use of internal controls. Were these genes amplified in every single reaction? Why not use a gene outside the system, especially considering KIR3DP1 might be absent in a few haplotypes?
- Clarification is also needed for HLA genotyping. What is the genotyping resolution that this method achieves for each locus? Does it discriminate HLA alleles? Does this method discriminate HLA-Bw4 from HLA-A and HLA-B? How about Bw4-80I from Bw4 80T?
- There are numerous problems with the haplotype inference performed in this study. 1) Although the method Haplo-IHP has been used in the past, this is a very limited method for haplotype inference based on presence and absence data. It is a consensus that precise haplotype inference without copy number of each gene and allelic information. 2) The limitation is even more critical for a study analyzing a highly diverse population and with relatively small sample size. 3) The method is based on a predefined haplotype list, and the list used was originated in a population from a completely different genetic background. In my opinion, these limitations are too strong to be ignored, to simply run the software, and list the results. The proportion of inaccurate haplotypes will be too high, and listing them does not contribute to the field. If the authors want to perform a haplotype inference, these limitations have to be crystal clear and there is no need to report the low-frequency (inaccurate) haplotypes. The software will likely be more accurate only for the highly frequent ones; therefore, reporting only those haplotypes is more responsible. The authors can look to the most frequent haplotypes in other South American populations and discuss the data responsibly, which would be much more impactful than a list of inaccurate haplotypes.
- The authors chose a list of 41 populations without apparent criteria. Why were these populations selected? What were the inclusion/exclusion criteria? Were the methods and sample sizes also taken into consideration? More importantly: why are these populations essential to compare with Colombia? I see a low representation of African and Native Americans, which is not understandable considering the background and geographic location of the population. What is the purpose of including so many Chinese and European-Descendant populations? The authors missed an opportunity to compare the study population with other South Americans, especially because many studies described their KIR variation. My suggestion is to carefully select a few references population representing major ancestral groups and include more populations from South America, including Native Americans, defining clear inclusion/exclusion criteria for both methodological and anthropological aspects.
- The authors must bear in mind that a part of the study population's differences compared to others might come from methodological differences. KIR variation is complex to genotype, particularly in the past. It means that there is much inaccurate data is out there. Finding criteria for comparison populations is critical (similar methods, accurate results, sample sizes, etc). For example, the linkage disequilibrium patterns of certain KIR are well known, and it is not easy to detect previously published data with questionable results just by looking at these patterns.
- In light of the previous comment, please be critical of the limitations of the genotyping method used for this study. Methods based on SSP and SSOP are prone to errors, even the commercial ones, due to the particularities of the system. Also, these methods were not designed to account for the allelic diversity of all populations, and it is likely to provide false results. One point that catches the attention is the high number of genotypes observed only in one individual. They are frequently very similar to a more common genotype. This observation could be an indication of genotyping issues. An acceptable amount of errors is OK; however, these can deviate the conclusions by inflating the diversity of the study population, especially the number of genotypes/haplotypes. It is critical to be aware of and also discuss the limitations of the method.
- Please check if the NPR3 genotypes are similar to common haplotypes and could be explained by an error of one or two genes. Please do a similar analysis for all haplotypes observed only once. Based on my experience, many of these are errors. Repeats and/or genotyping with a different method may be required.
- Please clarify the data used for NJ trees and PCA (presence/absence, haplotypes, genotypes)? Please use appropriate language to explain the PCA graph (lines 177-178).
- Table 1 and Table 2 present the genes listed in different orders. Please use the same order, considering the KIR haplotypes. Please list genotypes and haplotypes according to the frequency.
- Table 3 – KIR3DS1+Bw4 is not a functional pair. Several studies have failed to show that Bw4 binds to KIR3DS1. Please justify the inclusion of this pair.
- Table 4 I see no reason for writing A, B, C, D instead of the name of the genes. This notation just adds confusion.
- The authors use the word “phylogenetic” multiple times, referring to their NJ three, which is incorrect. Although this method was initially designed for phylogenetic analysis, this is not its purpose in this study. The clades do not imply a phylogenetic relationship among populations; instead, the tree only groups those populations with similar presence and absence data for KIR. Similar grouping is expected for the PCA, which, in this case, it is just a different way to calculate and visualize de groupings. Please, revise the language, state the limitations of the method, and discuss appropriately. Caution is required once grouping may also result from demographic and stochastic reasons.
- How does the HLA-KIR variation of the study population compare with others?
- Discussion/conclusion are overall very descriptive and sometimes repetitive.
Reviewer 2 Report
This manuscript describes KIR genotype and haplotype frequencies as well as distribution of their HLA ligands in Colombian population from Medellin (161 individuals). 37 different genotypes and 21 different haplotypes were found. Comparisons with published other populations in the world are presented. Remarkable differences in KIR frequencies were observed, even with Bogota population in the same country, mostly due to difference in KIR3DS1 more frequent in Medellin than in any other population.
The text is written in good English. KIR and HLA typing was performed using the LIFECODES typing kits. I am not aware of any publication using these kits, and the company does not provide sequencies of primers and probes nor characteristics of products detected. Therefore, the photos of gels should be provided by the Authors to show validity of their typings.
„For each KIR- and HLA-SSO typing at least 60 events were collected” – what does it mean? Each individual sample typing was repeated a least 60 times?! It is hard to believe!
For KIR2DS4, only the presence versus absence of the gene was detected. This gene exists in two variants: full length and 22-bp deletant which cannot be anchored at the cell surface and therefore does not function as a receptor for HLA-A3/A11. Thus, the KIR/ligand analysis, as presented in Table 3, does not make sense. In addition, the full length and deletant genes are distributed with very different frequencies in human populations, so it would be very interesting to see their proportion in the Medellin and Bogota populations.
The Bw4 epitope is present not only on many HLA-B allotypes, but also on some HLA-A allotypes (HLA-A*23, HLA-A*24, HLA-A*25 and HLA-A*32) as well. The Authors do not say whether their test detected Bw4 on HLA-B only or also on HLA-A.
KIR AA and Bx are genotypes, not haplotypes!
The „IndianN” and „IndianO” as well as „Iran” and „IranP”, appearing in Figures 3 and 5, should be explained. Similarly other abbreviations such as „USACa” and „UsaCa2”, „Ireland” and „Ireland2” and so on. The Authors should explicitly refer to Supplementary Table 1 here.
Why such a big difference between Medellin and Bogota was observed? Figure 3 shows larger gap between these two populations than between Bogota and Indian (i.e., Asian Indian, not American Indian!) populations. It may result from differences in KIR3DS1 gene frequencies, nevertheless it requires a comment. Again, without gel photo one cannot be sure that KIR3DS1 typing was correct, if its frequency is highly different from all other human populations (Supplementary Table 1).
„distinct expression patterns of KIR genes and haplotypes in this population.” – not expression but percent of positives! Gene expression means mRNA and protein level.
LD between KIR2DL2 and KIR2DS2 is weaker here than in Europeans, similarly as was described for Brazilians. However, as both 2DL2+2DS2- (H143) and 2DL2-2DS2+ (H2 and H4) haplotypes are present (Table 2), the net frequencies of both genes are almost identical in Medellin, but not in Bogota (Supplementary Table 1). This is the second factor differentiating between these two Colombian populations, and this should be also pointed out in Discussion.
In conclusion, the manuscript cannot be accepted and published in its present form. Results should be corrected for KIR2DS4 full length versus 22-bp deletion variants, Bw4 on HLA-B versus HLA-A, and other ambiguities listed above should be resolved.